# Multi-Parametric Diagnostic Approach and Potential Markers of Early Onset Subclinical Cardiovascular Disease in a Cohort of Children, Adolescents and Young Adults Vertically Infected with HIV on cART

**DOI:** 10.3390/jcm10225455

**Published:** 2021-11-22

**Authors:** Biagio Castaldi, Gloria Lanzoni, Osvalda Rampon, Daniele Donà, Angela Di Candia, Ornella Milanesi, Carlo Giaquinto, Giovanni Di Salvo

**Affiliations:** Department of Women’s and Children’s Health, University of Padua, 35126 Padua, Italy; gloria.lanzoni@icloud.com (G.L.); osvalda.rampon@aopd.veneto.it (O.R.); daniele.dona@unipd.it (D.D.); angeladicandia@me.com (A.D.C.); ornella.milanesi@unipd.it (O.M.); carlo.giaquinto@unipd.it (C.G.); giovanni.disalvo@unipd.it (G.D.S.)

**Keywords:** HIV, adolescents, cardiovascular risk, echocardiography, early-onset cardiovascular disease

## Abstract

Background: HIV infection and lifelong cART are responsible of an increase in cardiovascular risk. The aim of this study was to describe the subclinical cardiovascular disease and to identify early markers of cardiovascular damage in adolescents and young adults vertically infected with HIV on cART, through an innovative multi-parametric approach. Methods: We enrolled 52 patients vertically infected with HIV. Demographic records, traditional cardiovascular risk factors, laboratory findings and echocardiographic measurements were collected in a one-year routine follow up. The echocardiographic examination included measurements of the 2D and 3D ejection fraction (EF), E/A ratio, E/E′ ratio, carotid intima media thickness (cIMT), flow-mediated dilation (FMD) and global longitudinal strain (GLS). Results: At the time of enrolment, all the patients were on cART therapy. The viral load was suppressed in 95% of them. EF was normal in 94.2% of patients (66 ± 7.2%), and GLS (mean value: −20.0 ± 2.5%) was reduced in 29% of patients. The cIMT mean value was higher than the 95th centile for sex and age in 73%, and FMD was impaired in 45% of patients. Clinically evident disease was found in three patients: dilative cardiomyopathy in one, thoracic-abdominal aneurysm Crawford type II with a bilateral carotid dilation in one and carotid plaque with 30% of stenosis in a third patient. Conclusions: This study confirms the presence of clinical and subclinical cardiovascular disease in a very young population vertically infected with HIV, underlining the importance of an early, multi-parametric cardiovascular follow up.

## 1. Introduction

Although the antiretroviral therapies led to an increased survival of patients infected with human immunodeficiency virus (HIV), the population of children and adolescents vertically infected with HIV is at high risk for long-term non-acquired immune-deficiency syndrome (non-AIDS)-related complications, due both to chronic immune activation and early cellular senescence [1]. These processes lead to a lifelong systemic inflammatory condition, which is a driver for multi-systemic diseases.

Many studies highlighted an increased cardiovascular risk in HIV-infected adults while only a few research studies addressed on early identification of this damage among the young HIV-infected population [2]. Among the pediatric population, dilative cardiomyopathy and ventricular hypertrophy are the more often described cardiac pathologies [3,4,5].

The improvement of echocardiographic techniques allows for a more accurate study of the heart, by the measurement of strain and the strain rate through speckle tracking echocardiography (STE): longitudinal, circumferential and global strain were found to be more sensible and significantly altered in this particular population, when compared with ejection fraction and fractional shortening values [6,7].

Other markers of cardiovascular risk in HIV vertically infected patients are carotid intima media thickness (cIMT) and flow-mediated dilation (FMD) [8].

Some of the studies concerning heart disease in HIV-infected young people identified a correlation between cIMT values, virologic and immunologic markers (such as viral load, CD4+ lymphocytes) and antiretroviral therapy duration and regimens [9], but none of these studies included a full multi-parametric approach to the cardiovascular risk analysis through echocardiography.

The aim of this study was to evaluate a multi-parametric approach detecting cardiovascular and endothelial function by echocardiography in a cohort of pediatric and young adult patients vertically infected with HIV under effective antiretroviral therapy, in order to verify the prevalence of subclinical and clinically relevant cardiovascular disorders.

## 2. Method

This single-center, cross-sectional observational study was performed in the Department of Women’s and Children’s Health—University of Padua (Italy). All 52 children aged more than 5 years old, vertically infected with HIV and receiving antiretroviral therapy were consecutively enrolled in this study. Exclusion criteria were unknown way of transmission and not having vertically acquired HIV infection. All the patients and/or their parents signed a written consent to participate to this study, approved by the Ethical Committee for Clinical Studies of Padua.

During a routine follow-up visit, the patients underwent a clinal examination, blood sample analysis and the dedicated echocardiographic examination described below in detail. Smoke habits were detected without the presence of the parents. In addition to clinical and anthropometric data, the age at which the patients began the antiretroviral treatment, the duration of treatment (in years) and the therapy regimens at time of enrolment were recorded. Regimens including PIs, NRTIs, a combination of Lopinavir and Ritonavir, NNRTIs and integrase inhibitors were studied. The presence or absence of clinically evident lipodystrophy was evaluated by the clinician. Systolic and diastolic blood pressure centiles were calculated by nomograms adjusted for age and sex [10].

The laboratory findings were used to assess the metabolic profile of these patients, namely the fasting glucose levels and lipid profile. Blood samples were taken the same day of the echocardiographic examination, after a fasting period of at least 12 h.

The virological and immunological statuses were investigated by the Immunology and Molecular Diagnosis Unit of University of Padua. These parameters were used to monitor cART goals, such as viral load suppression, optimal values of CD4+ lymphocytes’ count and percentage and ideal CD4/CD8 ratio. HIV RNA (copies/mL) was assessed through the Roche Taqman machine. CD4+ lymphocyte number (cells/mm^3^), CD4+ lymphocyte percentage and CD4+ cells nadir were measured, as well as CD8+ lymphocyte number (cells/mm^3^) and percentage together with CD4/CD8 ratio.

The echocardiographic examination was performed by an expert cardiologist through GE Vivid 9 Echograph. The measurements and results were interpreted in accordance with current guidelines and/or previous studies.

M-mode and 2D images, with or without Doppler or Tissue Doppler, were acquired with a M5S probe, while 3D and multiple planes images were acquired with a 4V probe. Carotid intima media thickness and flow-mediated dilation were measured through an 11L probe.

The measurements performed during the investigation were as follows:Left ventricle and left atrium-aorta M-mode examination [11];Tricuspid Annular Plane Systolic Excursion (TAPSE) M-mode examination [12];Left ventricle volume and ejection fraction measured through biplane Simpson’s method [13];Left atrium volume through biplane area/length method [13];Right ventricle fractional area charge [14];Trans-mitral and trans-tricuspid Doppler to assess E, A, E/A and deceleration time (DT) [15];Tissue Doppler to assess E′ values taken from lateral mitral annulus, right ventricle septum and free wall [15];Epicardial adipose tissue (EAT) was measured during end-systole from the parasternal long axis view, slightly off axis [16];3D full volumes through a four beats acquisition, obtained online [17];Bilateral carotid IMT measurement, which is traditionally used as an early marker of cardiovascular pathology, was measured from the posterior right common carotid wall, once the machine recorded the mean value on a 15 mm long segment [18]. Based on the age of our population, cut-off values (<95th centile) were between 0.47 and 0.51 mm in males and 0.44 and 0.48 mm in females [ref];Flow-mediated dilatation, as a marker of endothelial dysfunction, was calculated after a 4 min ischemic period from brachial artery [19]. The cut-off used for peak diameter change was 9%;Global longitudinal strain is a very sensible and specific marker of subclinical myocardial dysfunction and was calculated after speckle tracking echocardiography with AFI software [12]. Based on the published age-related lower GLS normal limit, the cut-off used was −19.0% [ref x2].

### Statistical Analysis

The statistical analysis was performed using SPSS Software (v. 19.0). Continuous variables were expressed as means and standard deviations or median and quartiles (Q25/Q50/Q75), depending on their distribution. The normal distribution was verified by Shapiro–Wilk test. Qualitative data were compared using the Mantel–Haenszel test. Continuous variables were compared using unpaired *t*-test or the Mann–Whitney U-test. The correlations were studied by linear regression analysis. The null hypothesis was rejected for a *p*-value <0.05. The comparison of dichotomic variables were performed by using the Chi-square test and applying the Yates’ correction or Fisher’s exact test, when appropriate.

A multivariate analysis was performed based on the results of univariate analysis. Carotid intima media thickness, global longitudinal strain and flow-mediated dilatation were primary outcomes of the statistical analysis.

## 3. Results

The demographic characteristics and laboratory findings are illustrated in Table 1. Mean age of the cohort was 20.96 ± 5.69 years. The patients included in the study were 50% males. There were 15 patients of African origins (28.85%). There were 21 smokers (40.38%) at the time of enrolment.

Overall, 16/43 patients (37.2%) started cART when younger than 5 years old, while 27/43 patients (62.8%) started cART when they were older than 5 years old. Of the 52 patients enrolled, 9 patients (17.3%) started cART in another center, so the data regarding the beginning of treatment are missing.

The mean duration of treatment was 13.89 ± 5.16 years. Most of the regimens included PIs (71.15% of the cohort), while 40.38% of the patients were on Abacavir, 13.46% assumed a combination of Lopinavir and Ritonavir, 26.92% were on NNRTIs and 15.38% assumed integrase inhibitors.

The mean value of BMI was 21.94 ± 3.97 kg/m^2^. The mean value of systolic blood pressure was 121.20 ± 14.20 mmHg (range 155–96 mmHg), while the mean value of diastolic blood pressure was 74.63 ± 8.76 mmHg (range 97–51 mmHg). Lipodystrophy was present in 23.08% of the cohort. The total cholesterol mean value was 4.38 ± 0.96 mmol/L, triglycerides mean value was 1.25 ± 0.84 mmol/L and mean value of blood sugar was 4.71 ± 1.09 mmol/L.

HIV RNA was suppressed (<34 copies/mL) in 95% of the cohort. The mean value of CD4+ cells was 654.33 ± 273.52 cells/mm^3^, while the mean percentage was 31.21 ± 10.35%. The CD8+ lymphocytes mean value was 774.88 ± 332.46 cells/mm^3^ and the mean percentage was 37.08 ± 12.52%. The mean value for the CD4/CD8 ratio was 0.95 ± 0.47.

The echocardiographic findings are reported in Table 2. One patient suffered of chronic renal failure and presented a dilative cardiomyopathy with an EF of 42%. He received a kidney transplantation in March 2015. None of the remaining patients showed systolic dysfunction (EF 65.59 ± 4.88%; quartiles 62%/65%/69%), nor E/A ratio inversion (mean value 1.85; quartiles 1.7/1.8/2.2). Three patients (5.7%) presented an E/E′ ratio > 8.0 (mean value 5.7 ± 1.3; quartiles 4.5/5.5/6.3).

The mean value of GLS was −20.04 ± 2.52% (quartiles: −18.2%/−20.7%/−22%). The prevalence of pathologic modification in GLS value was 29% (Table 2).

The mean value of cIMT, normalized for age and sex, was 0.54 ± 0.12 mm (quartiles 0.49/0.53/0.54 mm). Given these results, cIMT was altered in 73% of the cohort, and this same percentage of patients presented a cIMT value higher than the 95th percentile for age and sex. Even when considered the Z-score of cIMT, the values were mostly over two standard deviations from normal value (Figure 1).

The mean value of FMD was 9.4 ± 4.7%, with a prevalence of FMD alteration of 45%.

Three patients (5.7%) presented clinically evident cardiovascular pathologies: one with a dilative cardiomyopathy, a second patient with an incidental diagnosis of thoracic-abdominal aortic aneurysm (Crawford type II) associated with a bilateral carotid dilatation and a third patient presented a carotid plaque that determined a 30% stenosis of carotid lumen.

Age and ethnicity of the patients, duration of antiretroviral therapy and CD4+ lymphocytes nadir data demonstrated a significant relation with cIMT (Table 3). In addition, patients with a higher cIMT Z-score were younger (16.68 ± 3.98 vs. 24.10 ± 4.63 years, *p* < 0.05) (Figure 1) and most of them were Black (63.64% vs. 3.33%).

The carotid IMT value was higher in younger patients (16.7 ± 4.0 vs. 24.1 ± 4.6 years, *p* < 0.001) and more frequently in patients who started therapy before the five years of life (*p* = 0.04). cIMT modifications neither appear to be significantly related to traditional cardiovascular risk factors (BMI, blood pressure, lipodystrophy, smoke habit and metabolic profile), nor to specific antiretroviral regimens or virological and immunological conditions. On the other hand, patients with higher cIMT had lower GLS values (Figure 2).

The univariate analysis pointed out a correlation between GLS value and CD4+ lymphocytes count: when a higher number of lymphocytes was present, the value of GLS tended to be normal (Figure 3).

FMD value presented a positive linear correlation with the CD4+ lymphocytes count, CD4+ percentage and CD4/CD8 ratio.

The epicardial adipose tissue (EAT) thickness was measured in 42 out of the 52 patients enrolled, with a mean value of 7.02 ± 2.36 mm (range 3–15 mm) during end-systole. Considering the cut-off proposed by Iacobellis et al. [16], epicardial fat thickness was higher than normal in 22% of the patients.

Overall, 83% of patients showed at least one positive marker of subclinical cardiovascular disease, 27% of patients had two positive markers and 29% of them showed three positive markers of subclinical cardiovascular impairment.

## 4. Discussion

Cardiac dysfunction has been previously reported in adults and children with HIV infection, although it is not clear whether those alterations are related to HIV or to antiretroviral treatment [20]. Despite clinical disease impact on a small part of patients, this study identifies a widespread distribution of subclinical cardiovascular alterations. On the other hand, the HIV infection was well controlled, given the viral load was suppressed in 95% of patients enrolled.

Previous studies evaluated the impact of lifelong antiretroviral therapy and non-AIDS-related cardiovascular disease, in particular by monitoring cIMT [20], which is a well-known and widespread marker of subclinical cardiovascular disease for the general population. It is obtained through a non-invasive technique and could also be reliable and validated to study a young HIV-infected population. Despite some conflicting data, the large part of these studies showed an increase in cIMT.

Accordingly, in our study, cIMT was higher than the 95th centile in 73% of patients. Subjects with cIMT Z-score >95th centile were younger, more often started cART protocol before the scholar age and most frequently were Black patients. These data suggest the presence of many variables impacting on this parameter: ethnicity, role of the antiretroviral therapy on the endothelium, changes of therapeutic protocols over the past decades, suboptimal therapeutic control in the first months of care (especially for poor social contests or due to linguistic barriers), etc. [21,22].

The laboratory exams do not show any alteration in metabolic profile. Some patients presented clinically evident lipodystrophy. In accordance to what is reported in the literature, therapy regimens that included PIs do not correlate with alterations in lipid profile, confirming that ARV therapies could lead to a better control of non-AIDS-related diseases [2]. The echocardiographic measurement of epicardial adipose tissue (EAT) thickness represents a novel marker of subclinical cardiovascular risk, as an increase in EAT thickness may identify individuals at higher likelihood of having detectable carotid atherosclerosis [23].

The echocardiographic examination was crucial to state the presence of subclinical or clinical cardiovascular disease. The distribution of clinical cardiovascular disease is in line with previous studies [6,24,25]. On the other hand, the magnitude of subclinical cardiovascular disease seems to be larger than previously reported.

In this study, for the first time, we used speckle tracking echocardiography to study cardiovascular mechanic in pediatric and young adult HIV-infected patients. Surprisingly, we found 29% of patients with GLS values below the lower normal limit (−19%). Similar data were recently found in adult HIV patients [26]. These findings could be due both to the effects of HIV infection itself on myocardial structure and vasculature and to the effects of chronic immune activation and early cellular senescence. Accordingly, some studies [27,28] demonstrated an involvement of micro-vessels through cardiac MRI. Interestingly, it was possible to point out a correlation between GLS and cIMT (r = 0.38, *p* = 0.03), indicating the presence of diffuse or multi-organ endothelial dysfunction (Figure 2). There are concerns that ART may induce atherosclerotic process and is associated with an increase in peripheral and coronary artery disease in these patients, leading to impairment in cardiac function [26]. The etiology of subclinical endothelial diseases might be multifactorial. Elevated levels of lipids are frequently reported in adult patients with HIV infection, in particular in patients on ART regimens [29]. Insulin resistance and diabetes are also another important issue in HIV-infected patients [30,31].

On the other hand, antiretroviral drugs toxicity, opportunistic infections, myocardial damage due to HIV infection itself or immunosuppression may also contribute to the myocardial dysfunction [32].

In our population, lipodystrophy was present in 23% of patients, while only one patient suffered of not well-controlled insulin-dependent diabetes and none presented with hypertension.

Flow-mediated dilatation is a marker of endothelial dysfunction and a predictive factor for the development of cardiovascular diseases [33]. Despite that FMD is often used in adults with cardiovascular risk factors, there is little evidence in the literature about routine assessments of cardiovascular risk in children and adolescents.

In the cluster of HIV-infected young patients, this kind of assessment is particularly helpful, given the higher risk of early onset of cardiovascular diseases. A case–control study performed on obese children and adolescents demonstrate the reliability of FMD in the determination of endothelial dysfunction [34]. These findings support the usage of FMD as a marker of early onset cardiovascular disease in children exposed to chronic inflammation. In our study, FMD was impaired in 45% of the patients enrolled, confirming the high risk of these patients. Between patients with a high cIMT, 54% of them also had a reduced GLS, while no patient with normal cIMT had reduced GLS (*p* = 0.01). On the other hand, 50% of patients with cIMT over the 95th centile also presented with an impaired FMD, compared to 14% in normal cIMT subjects (*p* = 0.02). When either cIMT or FMD were impaired, the percentage of reduced GLS rose to 77% (11% in the negative group, *p* < 0.0001; sensitivity 83%, specificity 84%), suggesting that the association of increased cIMT and increased FMD might be a marker of diffuse endothelial disease.

According to Kaplan et al. [35], a low CD4+ T-cell count is strongly related with an increased subclinical carotid atherosclerosis in HIV-infected adults, beyond traditional cardiovascular disease risk factors. Besides, Bernal E. et al. [36] noticed that the inversion of the CD4:CD8 ratio in HIV-infected patients on cART is independently associated with cIMT progression: this finding highlights a correlation between the cIMT value and conventional immunologic markers. Our study confirms that patients vertically infected with HIV on cART present a significantly increased risk of cardiovascular disease, which appears to be related to African ethnicity, therapy started in pre-scholar age and immunologic profile. The latter points out a correlation between the number of CD4+ cells and the value of GLS, while FMD shows a linear correlation with CD4+ cells number, percentage and with the CD4/CD8 ratio.

A multi-parametric approach is particularly important in these patients, because the cardiovascular involvement is already present in HIV+ pediatric patients, despite an effective control of the infection. All the echocardiographic measurements demonstrating a high rate of impairment (cIMT, FMD and GLS) should be routinely used in a well-structured follow up of these patients, being non-invasive, repeatable and well tolerated by the patients. On the other hand, the use of these cardiovascular parameters is far from able to be used in pediatric age patients, and tools and skills are not largely available. The early deterioration of endothelial function in these patients may lead to an early presentation of clinically evident cardiovascular diseases, and only an early diagnosis may allow for an appropriate and tempestive treatment of these complications. As a consequence, a dedicated and skilled team should be built for the follow up of vertically infected HIV patients.

This study suffers of some limitations. First, these data should be confirmed at a long-term follow up, to evaluate if these markers may have a predictive role on the onset of clinically relevant cardiovascular diseases. Second, as largely discussed, the etiology of the alterations in GLS, cIMT and FMD reported above can be multifactorial. Despite that the cohort enrolled was young, larger studies should be performed to weigh up the relative role of single factors on the endothelial dysfunction.

## 5. Conclusions

Despite the effective viral suppression, patients with vertically transmitted HIV infection are at high risk of early onset of cardiovascular diseases. If clinical disorders are relatively uncommon (5.7%), the 83% of the subjects enrolled in this study presented at least one marker of subclinical cardiovascular disease.

Therefore, a systematic multi-parametric and integrated approach should be considered in these patients.

Further studies should evaluate the prognostic role of these markers on the progression of cardiovascular disorders in HIV patients.

## Figures and Tables

**Figure 1 jcm-10-05455-f001:**
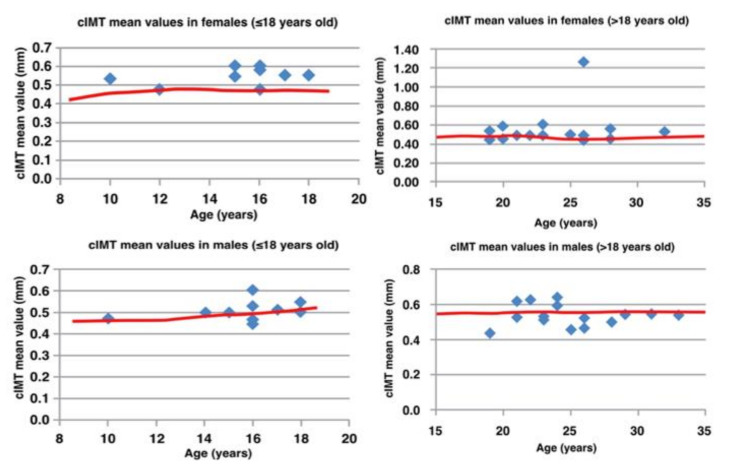
Distribution of cIMT values by age and sex. Values above the 95th centile for age and sex are reported above the red line.

**Figure 2 jcm-10-05455-f002:**
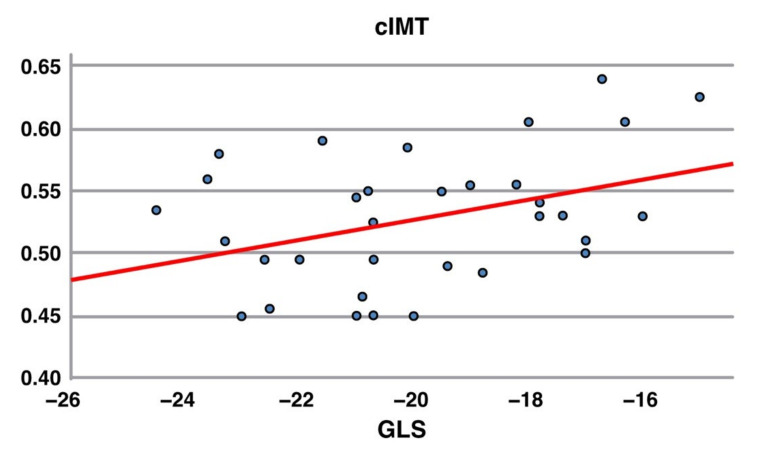
Linear correlation between cIMT values and GLS values. R = 0.38; *p* = 0.03.

**Figure 3 jcm-10-05455-f003:**
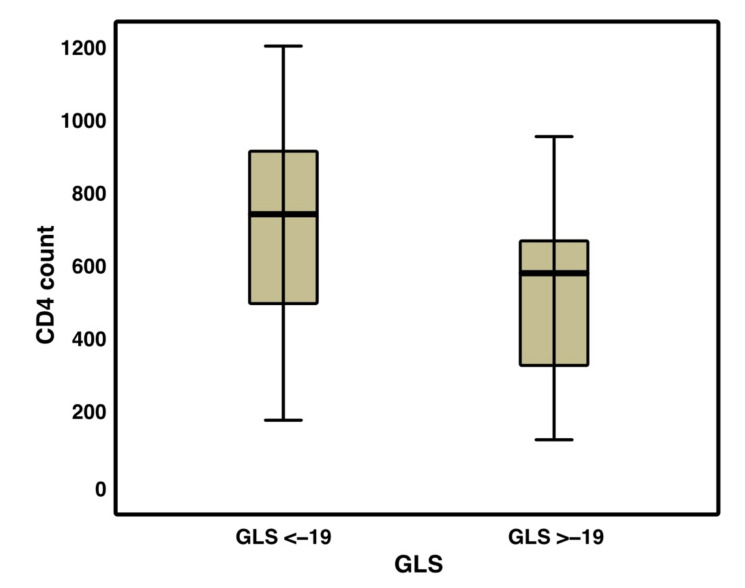
Correlation between GLS value and CD4+ lymphocytes count (cells/mm^3^) (*p* = 0.034).

**Table 1 jcm-10-05455-t001:** Descriptive statistics (sex, age, ethnicity, traditional cardiovascular risk factors assessment, virological and immunological status and therapy characteristics). Data are shown as number (#), (%) or mean (SD).

	*n* (%)	Mean (SD)
Male	26 (50%)	-
Age at examination (years)	-	20.96 (5.69)
Ethnicity #		-
E. Europe	4 (7.69%)	
W. Europe	33 (63.46%)	
Africa	15 (28.85%)	
Other	0 (0%)	
Current smoker	21 (40.38%)	-
Age ART initiated #		-
<5 years old	16 (37.2%)	
≥5 years old	27 (62.8%)	
Therapy duration (years)	-	13.89 (5.16)
PI therapy #	37 (71.15%)	-
Abacavir therapy #	21 (40.38%)	-
Kaletra therapy #	7 (13.46%)	-
NNRTI therapy #	14 (26.92%)	-
Integrase therapy #	8 (15.38%)	-
HIV RNA *#*/mL	-	0.29 (0.46)
CD4 nadir ***#*/**mm^3^	-	223.11 (229.51)
CD4 count ***#*/**mm^3^	-	654.33 (273.52)
CD4 % (%)	-	31.21 (10.35)
CD8 count ***#*/**mm^3^	-	774.88 (332.46)
CD8 % (%)	-	37.08 (12.52)
CD4/CD8 ratio	-	0.95 (0.47)
HDL cholesterol mmol/L	-	1.43 (0.40)
LDL cholesterol mmol/L	-	3.07 (0.95)
Total cholesterol mmol/L	-	4.38 (0.96)
Triglycerides mmol/L	-	1.25 (0.84)
Glycemia mmol/L	-	4.71 (1.09)
Mean blood pressure mmHg	-	90.51 (9.34)
BMI (kg/m^2^)	-	21.94 (3.97)
Lipodystrophy #	12 (23.08%)	-

**Table 2 jcm-10-05455-t002:** Echocardiographic values. Carotid intima media thickness (c IMT), global longitudinal strain (GLS), ejection fraction (EF) measured in 3D and 2D, E/E′ ratio and flow-mediated dilatation (FMD) were considered markers of clinical and subclinical cardiovascular disease, according to the American Heart Association Echocardiography Guidelines.

Measurements	Reference Values	Mean Value (SD)	Prevalence of Alteration (%)
c IMT	0.47–0.51 mm in males 0.44–0.48 mm in females	0.54 (0.12)	38 (73%)
GLS	<−19.0%	−20.04 (2.52)	15 (29%)
EF (2D)	≥55%	66.04 (7.37)	3 (5%)
EF (3D)	≥53%	65.59 (4.88)	0%
E/E′ ratio	<8%	7.15 (1.25)	4 (8%)
FMD	>9%	9.4 (4.7)	22 (45%)

**Table 3 jcm-10-05455-t003:** Univariate statistical analysis results. Primary outcome: prevalence of cIMT > 2 SD over non-normalized mean of 22 (42.31%). Secondary outcomes: prevalence of GLS 2SD over mean of 10 (19.23%) and prevalence of EF3D 2SD over mean of 15 (28.85%). Exploratory: linear regression of cIMT value with time on ART, key agent (separate models for PI, Abacavir, Kaletra, NNRTI and integrase), current smoker (yes/no), RNA copies, CD4 nadir and total cholesterol. Data are shown as number (#), (%) or mean (SD).

	Total	Normal cIMT (*n* = 30)	Elevated cIMT (*n* = 22)	*p* Value
Male, *n* (%)	26 (50%)	14 (46.67%)	12 (54.55%)	0.575
Age at examination (y)	20.96 (5.69)	24.1 (4.63)	16.68 (3.98)	<0.001
Ethnicity, #				<0.05
E. Europe	4 (7.69%)	4 (13.33%)	0 (0.00%)	
W. Europe	33 (63.46%)	25 (83.33%)	8 (36.36%)	
Africa	15 (28.85%)	1 (3.33%)	14 (63.64%)	
other	0 (0%)	-	-	
Current smoker,	21 (40.38%)	15 (50.00%)	6 (27.27%)	0.146
Age ART initiated, #				0.04
<5 years old	19 (36.54%)	7 (23.33%)	12 (54.55%)	
≥5 years old	33 (63.46%)	23 (76.67%)	10 (45.45%)	
Therapy duration (y)	13.89 (5.23)	16.44 (3.60)	10.67 (5.26)	<0.001
PI therapy #	37 (71.15%)	22 (73.33%)	15 (68.18%)	0.685
Abacavir therapy #	21 (40.38%)	9 (30.00%)	12 (54.55%)	0.075
Kaletra therapy #	7 (13.46%)	3 (10.00%)	4 (18.18%)	0.393
NNRTI therapy #	14 (26.92%)	8 (26.67%)	6 (27.27%)	0.961
Integrase therapy #	8 (15.38%)	5 (16.67%)	3 (13.64%)	0.765
HIV RNA copies/mL	0.29 (0.46)	0.34 (0.48)	0.23 (0.43)	0.366
CD4 nadir *#*/mm^3^	223.11 (229.51)	141.00 (150.16)	409.72 (274.02)	<0.05
CD4 count *#*/mm^3^	654.33 (273.52)	635.70 (296.13)	679.73 (243.77)	0.345
CD4 %	31.21 (10.35)	31.27 (11.66)	31.14 (8.51)	0.585
CD8 *#*/mm^3^	774.88 (332.46)	804.00 (398.65)	735.16 (214.72)	0.875
CD8 %	37.08 (12.52)	38.79 (14.68)	34.75 (8.57)	0.560
CD4/CD8 ratio	0.95 (0.47)	0.96 (0.54)	0.941 (0.358)	0.802
HDL cholesterol mmol/L	1.43 (0.40)	1.22 (0.34)	1.63 (0.35)	0.022
LDL cholesterol mmol/L	3.07 (0.95)	3.44 (1.07)	2.73 (0.72)	0.105
Total cholesterol mmol/L	4.38 (0.96)	4.37 (1.01)	4.40 (0.91)	0.875
Triglicerides mmol/L	1.25 (0.84)	1.28 (0.77)	1.20 (0.94)	0.225
Glycemia mmol/L	4.71 (1.09)	4.60 (0.47)	4.85 (1.59)	0.402
Mean blood pressure mmHg	90.51 (9.34)	92.59 (9.03)	87.27 (9.12)	0.079
BMI kg/m^2^	21.94 (3.97)	22.40 (4.09)	21.30 (3.81)	0.313
Lipodystrophy #	12 (23.08%)	10 (33.33%)	2 (9.09%)	0.040
cIMT mm	0.54 (0.12)	0.51 (0.05)	0.58 (0.16)	0.014
GLS *%*	−20.04 (2.52)	−20.33 (2.23)	−19.68 (2.85)	0.558
EF (2d) *%*	66.04 (7.37)	66.74 (6.43)	65.11 (8.53)	0.955
EF (3d) *%*	65.59 (4.88)	65.38 (4.67)	65.84 (5.29)	0.594
E/E′ ratio	7.15 (11.25)	5.52 (1.09)	9.30 (17.07)	0.403

## Data Availability

Data can be obtained following written request to the corresponding author.

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
