# Peer review of "Multi-Parametric Diagnostic Approach and Potential Markers of Early Onset Subclinical Cardiovascular Disease in a Cohort of Children, Adolescents and Young Adults Vertically Infected with HIV on cART"

_jcm, 2021, doi:10.3390/jcm10225455_

Round 1
Reviewer 1 Report
A very relevant study. My only concern is that there is no clear hypothesis. It could be an exploratory analysis. If it is an exploratory analysis then it should be clarified in the manuscript.
For example, the authors have checked markers for atherosclerosis such as intima thickness and marker of endothelial dysfunction FMD. If the aim was to check the process of atherosclerosis and endothelial dysfunction then there are other markers more specific for these issues.
Authors have also checked for echo findings including GLS. This checks different pathogenesis.
Author Response
Reviewer 1
A very relevant study. My only concern is that there is no clear hypothesis. It could be an exploratory analysis then it should be clarified in the manuscript.
Thank you for your comment. We tried to get more clear the aim of the study:
“The aim of this study was to evaluate a multi-parametric approach detecting cardiovascular and endothelial function by echocardiography in a cohort of pediatric and young adult patients vertically infected with HIV under effective antiretroviral therapy, in order to verify the prevalence of subclinical and clinically relevant cardiovascular disorders”.
For example, the authors have checked markers of atherosclerosis such as intima media thickness and marker of endothelial dysfunction FMD. If the aim was to check the process of atherosclerosis and endothelial dysfunction then there are other markers more specific for the issue.
We know that there are different markers and methods to study atherosclerosis, including diffenent imaging techniques and serological markers. However, we choose echographic parameters because non-invasive, cheap, repeatable, easily available and well tolerated.
Authors have also checked for echo findings including GLS. This checks different pathogenesis.
Thank you for this comment. We recognize that in HIV patients an impairment of GLS may be due to different factors, including macrovascular or microvascular alterations, infections, side effects of therapy, specific HIV-related damage, etc. However, irrespectively from the specific pathogenesis, our study suggests to check the trend of these values over the time to effectively and timely face an heart involvement. Despite we already stated this point in the limitations, we better clarified this point
“Second, as largely discussed, the aetiology of the alterations in GLS, cIMT and FMD reported above can be multifactorial.”
Reviewer 2 Report
This work is about the diagnostic approach and potential markers of early-onset subclinical cardiovascular disease in a cohort of children, adolescents, and young adults vertically infected with HIV on cART. The author stated the incidence of cardiovascular disease was higher in this population. The author still needs to compare with the healthy population.
The conclusion is the 83% of the subjects enrolled in this study presented at least one marker of subclinical cardiovascular disease. The author needs to highlight this in the results.
The duration of treatment is not related to cIMT in Table 3. Could the author discuss more for this finding?
CD4 nadir showed the difference in Table 3. CD4 count seems to influence GLS. The p-value of the two bars in Figure 3 need to present.
Author Response
Reviewer 2
This work is about the diagnostic approach and potential markers of early-onset subclinical cardiovascular disease in a cohort of children, adolescents, and young adults vertically infected with HIV on cART. The author stated the incidence of cardiovascular disease was higher in this population. The author still needs to compare with the healthy population.
Thank you for this comment. The increased incidence of subclinical cardiovascular disease was analyzed considering reference values for markers of subclinical cardiovascular diseases for the age and sex. In particular, IMT and GLS have well defined normal cutoff values obtained by very large studies. As a consequence, the enrollment of a small healthy control cohort, in our opinion, would not change the values of the results shown with these parameters.
The conclusion is the 83% of the subjects enrolled in this study presented at least one marker of subclinical cardiovascular disease. The author needs to highlight this in the results.
Thank you for your suggestion. We added this data in the results section.
The duration of treatment is not related to cIMT in Table 3. Could the author discuss more for this finding?
This is a very good point. We faced this point in the 3rd paragraph of the discussion section.
“Accordingly, in our study, cIMT was higher than 95th centile in 73% of patients. Subjects with cIMT Z-score >95th centile were younger, more often started cART protocol before the scholar age and most frequently African black patients. These data suggest the presence of many variables impacting on this parameter: ethnicity, role of the antiretroviral therapy on the endothelium, changes of therapeutic protocols over the past decades, sub-optimal therapeutic control in the first months of care (expecially for poor social contests or due to linguistic barriers), etc [21, 22].”
CD4 nadir showed the difference in Table 3.
Thank you for this comment. As reported above, the first presentation of these patients was quite heterogeneous. We found a lower CD4 nadir values in older patients and in patients starting therapy in older age. On the other hand, after the first months of thaking charge, therapeutic regimen was effective and well controlled in all the patients.
CD4 count seems to influence GLS. The p-value of the two bars in Figure 3 need to present.
Thank you for this suggestion. We added p value (0.034)
Round 2
Reviewer 2 Report
The work improved after being revised. The author gave a good response to all comments.